# Characterization of Dosage Levels for In Ovo Administration of Innate Immune Stimulants for Prevention of Yolk Sac Infection in Chicks

**DOI:** 10.3390/vetsci9050203

**Published:** 2022-04-22

**Authors:** Mishal Sarfraz, Thuy Thi Thu Nguyen, Colette Wheler, Wolfgang Köster, Volker Gerdts, Arshud Dar

**Affiliations:** Vaccine and Infectious Disease Organization (VIDO), University of Saskatchewan, Saskatoon, SK S7N 5E3, Canada; mishal.sarfraz@usask.ca (M.S.); colette.wheler@usask.ca (C.W.); volker.gerdts@usask.ca (V.G.); arshud.dar@usask.ca (A.D.)

**Keywords:** CpG, poly I:C, innate immune stimulants, yolk sac infection, chickens

## Abstract

Innate immune stimulants, especially toll-like receptor (TLR) ligands and agonists, are the main players in the initiation of innate immunity and have been widely studied as alternatives to antibiotics to control infection. In the present study, we characterized the dosage levels of various innate immune stimulants, including unmethylated cytosine-phosphate-guanosine dinucleotide -containing oligodeoxynucleotides (CpG ODN), polyinosinic-polycytidylic acid (poly I:C), cyclic polyphosphazene 75B (CPZ75B), avian beta-defensin 2 (ABD2), and combinations of these reagents given in ovo. Data derived from a series of animal experiments demonstrated that the in ovo administration of 10–50 µg CpG ODN/embryo (on embryonic day 18) is an effective formulation for control of yolk sac infection (YSI) due to avian pathogenic *Escherichia coli* (*E. coli*) in young chicks. Amongst the different combinations of innate immune stimulants, the in ovo administration of CpG ODN 10 µg in combination with 15 µg of poly I:C was the most effective combination, offering 100% protection from YSI. It is expected that the introduction of these reagents to management practices at the hatchery level may serve as a potential replacement for antibiotics for the reduction of early chick mortality (ECM) due to YSI/colibacillosis.

## 1. Introduction

In the last few decades, the poultry industry has witnessed extraordinary production gains. For instance, in 1925, a 1.1 kg chicken was produced in 112 days, whereas, in 1950, the same weight was attainable in 70 days. In 2015, the production of a 2.27 kg chicken was possible in less than 50 days [1]. These gains were possible through the adaptation of intensive farming of genetically homogenous animals bred for high productivity traits and better disease control practices, including the preventive and therapeutic use of antibiotics [2]. Disease control in poultry production bears serious economic and social implications [3]. However, poultry infectious disease management is becoming increasingly difficult as regulatory changes are implemented to reduce environmental hazards, pathogen evolution, and food safety [4]. For example, legislation regarding the withdrawal of antibiotic use in poultry production has led to the development and characterization of effective, safe, and economical alternative strategies. These alternatives include innate immune stimulants, prebiotics, probiotics, enzymes, organic acids, the introduction of new vaccines, and the development of disease-resistant poultry breeds. Similarly, animal welfare issues driving the industry toward cage-free farming are leading to significant changes in the pathogen spectrum and new challenges for disease control [4,5,6].

Bacterial infections, including yolk sac infection (YSI) and colibacillosis (prime cause of early chick mortalities (ECMs)), clinical and subclinical *Clostridial* infection (causing necrotic enteritis), and Salmonellosis or asymptomatic colonization of *Salmonella* and *Campylobacter* play a critical role in profitability, food security, and food safety issues associated with poultry production [3,7,8,9]. For years, prophylactic antibiotics have been used for the prevention or control of these bacterial infections. However, the link between the use of antibiotics and the emergence of antibiotic-resistant pathogens has led to serious public health concerns about the continued use of antibiotics in poultry production [10]. For instance, Webster (2009) showed that the administration of ceftiofur (a third-generation cephalosporin) at 0.17 mg/chick subcutaneously or 0.08–0.20 mg/embryo via the in ovo route for the prevention of YSI and ECM in young broilers is strongly associated with the spread of cephalosporin-resistant, difficult-to-treat *E. coli* and *Salmonella* infections in Canadian human populations [11]. Interestingly, this author also indicated that a two-year voluntary ban on the use of cephalosporins by Québec hatcheries has greatly improved the efficacy of cephalosporins against *E. coli* and *Salmonella* infections in humans [11]. Similarly, over 20% of *E. coli* strains isolated from broilers in the Netherlands were resistant to cefotaxime (also a third-generation cephalosporin) in 2007. However, this resistance prevalence decreased sharply and reached a level of 2.9% in 2014 after the ban on the use of ceftiofur in hatcheries in 2010 [12]. Another study from the Netherlands found that extended-spectrum beta-lactamase (ESBL)-producing *E. coli* isolated from retail chicken meat samples and human clinical samples shared approximately 39% similarity, implying that ESBL genes could be transmitted from poultry to humans through the food chain [13]. However, these findings are contradicted by the lower prevalence of ESBL-producing *E. coli* in Sweden, as only 0.09% of the population might be expected to carry poultry-associated isolates [14]. The low quantity of antibiotics used in poultry in Sweden compared with that in the Netherlands is considered one of the major reasons for this remarkable difference in the dissemination of ESBL-producing *E. coli* between the Netherlands and Sweden [14]. These examples suggest that continued use, misuse, or overuse of antibiotics in food animal production, including poultry, is culminating in the emergence and circulation of multiple-drug-resistant pathogens (superbugs) in human and animal populations. Importantly, this grave risk of spreading antimicrobial resistance is coupled with all segments of the poultry industry, including farming, transportation of birds, manure handling, veterinary medicine, meat processing, and meat consumption [15]. Thus, in order to save antibiotics as an effective treatment option for public health, there is an urgent need to look for non-antibiotic-based alternatives for growth promotion and the prevention or treatment of bacterial infections in poultry.

Omphalitis, or yolk sac infection (YSI), in young chicks, is the major cause of ECM, with most death occurring within 24 h of hatching and peaking at 5 to 7 days [16]. These infections may cause up to 1.3% mortality per week during the first two weeks, uniformity in size, and greater susceptibility to other infectious diseases in young birds [17]. Previous studies demonstrated that the in ovo administration of a single dosage level of innate immune stimulants, including unmethylated cytosine-phosphate-guanosine dinucleotide -containing oligodeoxynucleotides (CpG ODN), polyinosinic-polycytidylic acid (poly I:C), and cyclic polyphosphazene 75B (CPZ75B), is safe and effective in the prevention of ECM due to YSI caused by avian pathogenic *E. coli* (*APEC)* [17]. In ovo immunization is a feasible, effective, less expensive, and more precise way to offer a uniform and appropriable level of protection. In ovo vaccination machines are currently utilized in over 90% of USA broiler hatcheries, and their use is expanding rapidly in Europe and Latin America as an efficient method to protect chickens from Marek’s disease, infectious bursal disease virus (IBDV), and poxvirus [18]. In this study, in ovo administration was achieved by puncturing a small hole through the blunt end of the eggs using an oblique needle and then passing down a smaller needle to deliver innate immunostimulants to the amniotic fluid of eggs on day 18 of incubation. Different dosage levels of CpG ODN, poly I:C, CPZ75B, avian beta-defensin 2 (ABD2), and their combinations were tested to determine the optimal dosage levels for in ovo use. These formulations were characterized with respect to their protective efficacy against ECM due to experimental YSI in young chicks caused by *APEC*. It is expected that the introduction of these reagents at the hatchery level will be helpful in reducing ECM, enhancing profitability, and reducing biological, environmental, and ecological hazards associated with the use of antibiotics in young chicks. 

## 2. Materials and Methods

### 2.1. Ethics Approval

All animal trials were conducted according to animal protocols (AUP 20160079) approved by the Animal Care Committee of the University of Saskatchewan.

### 2.2. Animals

In all experiments, specific-pathogen-free (SPF) eggs (Sunrise Farms Inc., Catskill, NY, USA) were incubated at the Poultry Science hatchery of the University of Saskatchewan. To ensure that the embryos were free of *E. coli*, yolk, liver, and heart tissue samples were taken from 5 randomly selected embryos on embryonic day (ED) 18 prior to the administration of innate immune stimulants and cultured on blood agar (Hardy Diagnostics, Santa Maria, CA, USA) and MacConkey agar (BD, Sparks, MD, USA) plates. Eighteen-day-old embryonated eggs were injected with 100 µL of test formulations or sham (PBS) solutions in ovo into the amniotic sac. Right after hatching, the chicks were transferred to the animal care unit of the Vaccine and Infectious Disease Organization (VIDO), University of Saskatchewan, for the challenge studies. All hatched birds were administered antibiotic-free feed and water ad libitum. 

### 2.3. Trial 1: Optimization of In Ovo Dosage Levels of Stand-Alone Innate Immune Stimulants

In this animal trial, multiple dosage levels of select innate immune stimulants were characterized in ovo. The treatment groups (n = 20 embryos/group) included CpG ODN at 50, 20, 10, 5, and 1 µg per embryo; cyclic polyphosphazene 75B (CPZ75B) (developed at VIDO) at 10 µg and 20 µg per embryo; and poly I:C at 5, 10, and 20 µg per embryo. The control groups received 100 µL PBS/embryo. 

### 2.4. Trial 2: Comparison of Single Use and Combinations of Innate Immune Stimulants

This trial was conducted to evaluate formulations prepared from stand-alone or combinations of CpG ODN, poly I:C, avian beta-defensin 2 (ABD2, synthesized by VIDO), and CPZ75B. All formulations and controls were administered in ovo in a volume of 100 µL/embryo. Briefly, on ED18, a total of 350 viable SPF chicken embryos were divided into seven groups designated as A to G comprised of 50 embryos (n = 50) per group. Stand-alone immune stimulants, including CpG ODN 20 µg/embryo, poly I:C 20 µg/embryo, and ABD2 10 µg/embryo, were administered to groups A, B, and C, respectively. Formulations containing combinations of immune stimulants included (Group D) CpG ODN 10µg + poly I:C 15 µg, (Group E) poly I:C 15 µg + ABD2 10 µg, and (Group F) CpG ODN 10µg + poly I:C 15 µg + ABD2 10 µg per embryo. Group G was injected with 100 µL of sterile PBS as a sham control.

### 2.5. Trial 3: Further Study for Combinations of Innate Immune Stimulants 

In this experiment, additional combinations of innate immune stimulants were tested. On ED18, 120 viable SPF chicken embryos were randomly divided into 4 equal groups (n = 30/group) designated as A to D. Groups A, B, and C were given formulations including (i) CpG ODN 20 µg + CPZ75B 10 µg, (ii) poly I:C 10 µg + CPZ75B 10 µg, or (iii) CpG ODN 20 µg + poly I:C 10µg + CPZ75B 10 µg/embryo, whereas group D was treated with 100 µL sterile PBS. 

### 2.6. E. coli Challenge 

The efficacy of each formulation was evaluated through the reduction in ECM for 7 days following a severe challenge using avian pathogenic *E. coli* strain EC317. *E. coli* EC317 was originally isolated from a case of septicemia in a turkey. This *APEC* strain was found to be highly pathogenic and causes both sepsis and cellulitis in chickens. *E. coli* EC317 belongs to serotype O2, has a K1 capsule and Type-1 pili, produces aerobactin, and is serum-resistant [19]. Birds were challenged following a protocol described earlier [17]. In brief, 25–30 CFU/100 µL PBS/chick were administered via the intra-navel route in 1-day-old chicks. Following the virulent challenge, all animals were assigned clinical scores and monitored for mortalities four times a day for 7 days. The clinical scores were recorded as follows: score 0 for normal birds; score 0.5 indicates birds with slow movement; score 1 for birds with ruffled feathers, sitting, hesitant to stand, and mouth breathing; score 2 for birds unable to stand or walk, unable to reach feed or water, wings extended, and exhibiting trouble breathing; score 3 for birds found dead. Birds having a clinical score of 2 were humanly euthanized. Dead and euthanized birds were necropsied and tissues, such as the yolk sac, liver, and pericardial sac, were swabbed and cultured on MacConkey plates for the isolation and identification of pathogenic *E. coli*. The daily survival rate was determined as the ratio of the number of alive birds in a specific group on that day to the number of challenged birds in that group.

### 2.7. Statistical Analysis

The survival curves from the treatment and control (PBS) groups were statistically analyzed using GraphPad PRISMTM 8 software (GraphPad Software Inc., San Diego, CA, USA). The data were normalized by log ranking. Statistical differences were determined by one-way analysis of variance (nonparametric, Kruskal–Wallis) and Chi-square tests (Gehan–Breslow–Wilcoxon). The differences were considered statistically significant at *p* < 0.05.

## 3. Results

### 3.1. Optimal Dosage Levels of Stand-Alone Innate Immune Stimulants

In order to determine the optimal dosage levels of innate immune stimulants used in our previous and current studies, multiple dosage levels of each innate immune stimulant were administered in ovo to respective groups of embryos. The efficacy of each dosage was determined through the evaluation of protection from ECM due to experimental YSI (caused by intra-navel challenge of 1-day-old birds with virulent *APEC*) in trial 1. For CpG ODN, our data showed survival rates of 76%, 81%, 77%, 60%, and 44% on the seventh day post-challenge following the in ovo administration of 50 µg, 20 µg, 10 µg, 5 µg, and 1 µg, respectively. There were no significant differences between the dosage levels of 50 µg, 20 µg, and 10 µg of CpG ODN. Survival rates at dosage levels of 5 µg and 1 µg were not different from the controls (Figure 1a, Table 1). Poly I:C treatment with 20 µg showed significantly higher survival rates (82%) compared with the other dosages and controls (Figure 1b, Table 1). Regarding cyclic polyphosphazene (CPZ75B), treatment with 20 µg and 10 µg showed survival rates of 60% and 39%, respectively, which were not significantly different from those of the control group (Figure 1c, Table 1). 

### 3.2. Protection of Combinations of Innate Immune Stimulants

Trial 2 showed survival rates of 77%, 82%, and 84% at 7 days post-challenge (PC) following the in ovo administration of CpG ODN 20 µg/embryo, poly I:C 20 µg/embryo, and ABD2 10 µg/embryo, respectively (Figure 2a, Table 2). These survival proportions were significantly higher than survival in the control (PBS) group (with 59% survival proportion). Amongst the groups of embryos treated with combinations of immune stimulants, the group treated with CpG ODN 10 µg/embryo + poly I:C 15 µg/embryo showed 100% survival in the first 7 days post-challenge, whereas embryos in groups treated with the poly I:C 15 µg/embryo + ABD2 10 µg/embryo or CpG ODN 10 µg/embryo + poly I:C 15 µg/embryo + ABD2 10 µg/embryo combinations showed survival proportions of 79% and 70%, respectively (Figure 2b, Table 2). However, statistically, there were non-significant differences between the groups treated with a stand-alone immune stimulant or the groups treated with combinations of immune stimulants. 

More combinations of immune stimulants, including (group A) CpG ODN 20 µg in combination with CPZ75B 10 µg/embryo, (group B) poly I:C 10 µg plus CPZ75B 10 µg/embryo, and (group C) a combination of three immune stimulants consisting of CpG ODN 20 µg plus poly I:C 10 µg plus CPZ75B 10 µg/embryo, were tested in trial 3. In these studies, groups A and C showed significantly higher survival rates (82 and 85%, respectively) compared with the control group at 62%, whereas group B showed survival at 60% which was not statistically different from that of the control group at 7 days PC (Figure 2c, Table 3).

## 4. Discussion

The use of antibiotics in food animal production is one of the major sources of the emergence of antimicrobial resistance (AMR) in animals and humans. AMR causes ~700,000 deaths per year worldwide [20,21]. Antibiotics in poultry production are used for multiple reasons, including the treatment of infections, prevention of the colonization of foodborne pathogens, and as growth promoters [22]. Public health, animal health, and animal welfare issues associated with the use of antibiotics in poultry have attracted global attention toward the development and evaluation of non-antibiotic-based alternative strategies for the prevention of these problems [23]. Importantly, the discovery of the crucial role of innate immunity in preventing infections and determining the nature and duration of adaptive immunity in early post-hatch life has opened new avenues to address the issues associated with the use of antibiotics in food animal production. Rapid progress in the identification of receptors, signaling pathways, effector molecules, and understanding the mechanisms of action of innate immune stimulants capable of offering protection from bacterial and viral infections has greatly helped in the development of novel and effective innate immunity-based strategies for protection from viral and bacterial infections in animals and humans [24,25,26,27]. Many researchers have shown that a stronger innate cytokine response following toll-like receptor (TLR) agonist stimulation in neonates compared with adults makes innate immune stimulants an attractive choice for the prevention of infections [28,29,30]. A number of reports have shown increased protection and reduced severity and duration of infection caused by pathogens, such as *Mycobacterium tuberculosis*, *Listeria*, and *Francisella* species, following the administration of CpG ODN in mice [31,32,33,34]. In addition, Thatte et al. (2011) and Ashkar et al. (2004) showed protection from herpes virus 2 virus infection in mice mediated through the activation of interleukin (IL)-15 following the administration of CpG ODN and poly I:C [35,36]. Similarly, the in vivo immune-protective effect of CpG ODN in a number of viral and bacterial infections in chickens has also been reported in multiple research reports [26,37,38,39,40]. Recently, Goonewardna et al. (2021) showed that the aerosolized delivery of CpG ODN in hatched chicks in a commercial hatchery setting offered protection from mortality and septicemia caused by lethal *E. coli* infection [41]. 

Poly I:C, another innate immune stimulant investigated in these studies, mediates innate immune activation through its binding to multiple dsRNA sensors, including retinoic acid-inducible gene I (RIG1), melanoma differentiation-associated protein 5 (MDA5), and TLR3 [42]. TLR3 and MDA5 signaling leads to the induction of interferon regulatory factor (IRF) 3 and NFĸb, resulting in type-I and type-III interferon (IFN) and proinflammatory chemokine production, whereas, type-I IFN activates neighboring IFN regulatory genes [43]. Primarily, the anti-infection role of poly I:C is mediated through the activation of cytokines and chemokines, including IFN-γ, IFN-β, and CXC chemokines (CXCL 9, 10, and 11). The activation of these proteins plays a significant role in the activation, differentiation (from naive T cells to Th1 cells), and migration of immune cells, including macrophages, cytotoxic lymphocytes (CTLs), natural killer (NK) cells, and natural killer T (NKT) cells [44]. It has been shown that activation of IFN-β and myeloid differentiation primary response 88 (MYD88) and the toll/interleukin1 receptor domain containing an adapter inducing IFN-β (TRIF) following poly I:C treatment in mice offers protection from *Yersinia enterocolitica*. Lanteir et al. (2014) showed that protection from *Cryptosporidium* and *Salmonella* Typhimurium infection following the administration of poly I:C in neonatal mice critically depends on IL-12p40, IFN-α, and IFN-β induction mediated through the activation of MDA5 and TRIF via TLR3 signaling [45]. Similar to these findings, Karpala et al. (2008) showed increased expression of TLR3 and type-I IFN in chicken leukocytes following poly I:C treatment [46]. In conjunction with the above reports, we have previously shown significantly reduced ECM and lower clinical scores due to experimental YSI in a group of birds treated in ovo with CpG ODN 50 µg/embryo [17]. Data from the present studies support and broaden our previous findings. We observed a significant reduction in ECM due to *APEC*-based YSI through the in ovo administration of 10–50 µg CpG ODN; meanwhile, using a dose of 5 µg/embryo or lower exhibited similar levels of protection to the PBS group. No differences in survival were found among groups treated with CpG ODN at 10, 20, and 50 µg/embryo, suggesting that the cost-effective dosage levels for the stand-alone use of CpG ODN delivered in ovo are between 10 and 20 µg/embryo (Trial 1). Additionally, birds receiving 20 µg of poly I:C or 10 µg of ABD2/embryo alone had survival rates comparable to those of birds treated with 20 µg of CpG ODN (Trial 2). Among the three innate immune stimulants (CpG ODN, poly I:C, and ABD2), CpG ODN is more expensive than poly I:C or ABD2. Hence, in order to apply these immunostimulants in the commercial poultry sector, the co-administration of CpG and poly I:C or ABD2 using a half-dose each may be beneficial in lowering the cost and enhancing *APEC* protection in young birds. 

Several reports have shown that the co-stimulation of multiple TLRs with selective synthetic TLR ligands may activate different TLR-associated adaptor proteins and signaling cascade-modulating factors leading to the augmentation or suppression of immune responses [47,48]. For instance, He et al. (2007) showed immune synergy of CpG ODN and poly I:C through the induction of enhanced antiviral and antibacterial activity in chicken primary monocytes treated with a combination of these two reagents. This in vitro synergetic effect was monitored through the induction of nitric oxide (NO) synthesis in chicken monocytes. Interestingly, the combination of CpG ODN/poly I:C with other innate immune stimulants (targeting different TLRs), including Pam3CSK (synthetic lipoprotein), peptidoglycan (PGN), lipoteichoic acid, lipopolysaccharide, flagellin, and loxoribine, showed no synergetic effect on NO production [49]. Likewise, Muvva et al. (2021) showed a beneficial effect of triggering multiple innate immune pathways by using multiple innate immune stimulants for the treatment of multidrug-resistant tuberculosis in human patients [50]. Bashir et al. (2019) reported the use of a combination of TLR agonists, including pam3CSK4 (TLR2 agonist) and poly I:C (TLR3 agonist), as an effective strategy to alleviate virus-induced immune suppression associated with the use of live attenuated infectious bursal disease vaccines (hot vaccines that may induce immune suppression) in chickens. This reduced immune suppression is attributed to the upregulation of IFN-β, IFN-γ, IL-12, and IL-4, and inhibition of IL-1β, IL-10, and inducible nitric oxide synthases (iNOS) expression through the synergistic effect of both TLR ligands [51]. Interestingly, in our studies, the group of embryos treated with CpG ODN 10 µg (half of the stand-alone dosage) in combination with 15 µg of poly I:C/embryo showed 100% survival in the first 7 days post-challenge (Figure 2b), whereas the stand-alone treatment of CpG ODN at 10 µg or poly I:C at 10 or 20 µg per embryo was not as effective as the combination. Additionally, our data (trial 2 and 3) showed comparable protection levels in the groups treated with CpG ODN 20 µg + CPZ75B 10 µg (82% survival), poly I:C 15 µg + ABD2 10 µg (79% survival), or CpG ODN 20 µg + poly I:C 10 µg + CPZ75B 10 µg (85% survival) with a stand-alone treatment of CpG ODN 20 µg/embryo (survival rate 81%). Data from both studies collectively indicate that there are comparable choices that may be selected on the basis of their availability and cost. For example, our data suggest that the use of avian beta-defensin in place of CpG ODN may be more economical for the prevention and control of ECM/YSI. However, further characterization is warranted for adaptation in the field.

Innate immune stimulants, such as CpG ODN and poly I:C, are not only potential anti-infection reagents, but also are good adjuvants to enhance immune responses for their incorporated vaccines. These immunostimulatory reagents have been intensively studied as adjuvants in poultry for improving both humoral and cellular immune responses. For instance, the in ovo administration of encapsulated CpG ODNs 2007 with herpesvirus of turkey (HVT) vaccine could moderately enhance HVT efficacy and reduce tumor incidence [52]. CpG ODN, as an adjuvant, could protect chickens from H9N2 avian influenza [53]. Regarding poly I:C, it was reported that the administration of poly I:C as an adjuvant in an H9N2 influenza vaccine significantly induced higher antibody titers and elevated expression levels of IFN-α, IFN-γ, IL-6, and major histocompatibility complex class II (MHC-II) in ducks [54]. In addition, poly I:C combined with gp90 recombinant protein from reticuloendotheliosis virus (REV) significantly reduced the viremia and immunosuppressive effects caused by REV challenge in chickens [55]. Moreover, a poly I:C and CpG ODN combination adjuvant enhanced antibody-mediated and cell-mediated immune responses against avian influenza viruses (AIV) antigens [56]. Therefore, further studies focusing on the co-delivery of these innate immune stimulants with in ovo vaccines, such as Marek’s vaccine, Newcastle disease, fowl pox, and coccidiosis, may be beneficial for the enhancement of vaccine effectiveness.

## 5. Conclusions

We have demonstrated that the in ovo co-administration of CpG 10 µg/embryo and poly I:C 15 µg/embryo provided the highest degree of protection against YSI (100% survival) in young birds. In addition, stand-alone use via the in ovo route of CpG ODN 10–50 µg/embryo, poly I:C 20 µg/embryo, or ABD2 10 µg/embryo could significantly reduce ECM due to severe YSI, with survival ranging from 76 to 84%. Similarly, the synergetic effects of various combinations of innate immune stimulants, including: (i) CpG ODN 20 µg + poly I:C 10 µg + CPZ75B 10 µg (85% survival), (ii) CpG ODN 20 µg + CPZ75B 10 µg (82% survival), or (iii) poly I:C 15 µg + ABD2 10 µg (79% survival) showed significant improvements in the survival rate following severe YSI. Further characterization of these formulations is warranted to investigate their effectiveness against other bacterial infections in young chicks.

## Figures and Tables

**Figure 1 vetsci-09-00203-f001:**
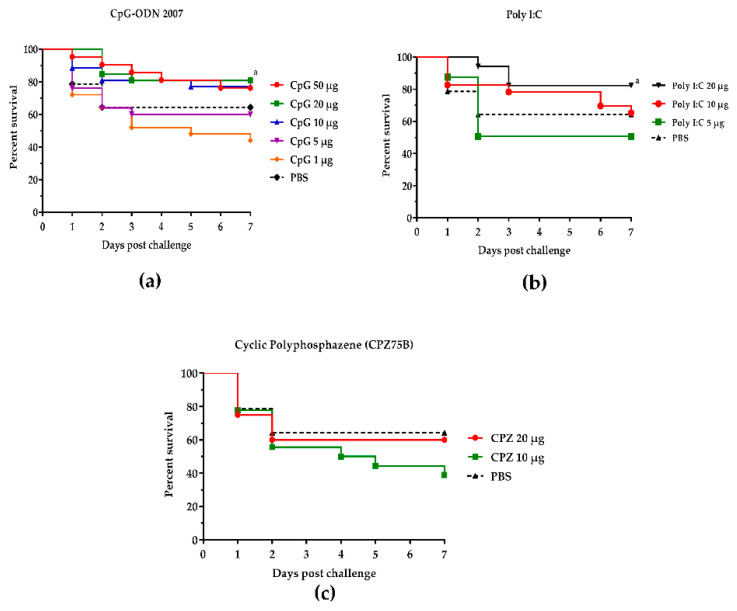
Protective effectiveness of different dosage levels of innate immune stimulants against YSI. (**a**) Dosage levels of CpG ODN 2007. (**b**) Dosage levels of poly I:C. (**c**) Dosage levels of CPZ75B. Lower case letters in the figures indicate a significant difference in the survival rates at *p* < 0.05.

**Figure 2 vetsci-09-00203-f002:**
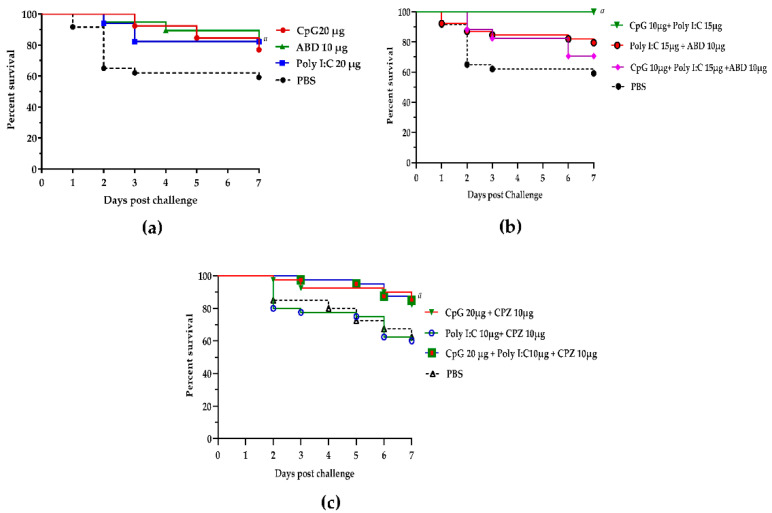
Protection of in ovo administration of innate immune stimulants alone and in combination. Statistical differences in the survival proportion were considered significant at *p* < 0.05 and are indicated as a different letter (the lowercase letter—a). (**a**) Survival proportions following stand-alone administration of CpG ODN 2007, poly I:C, and ABD2 (trial 2). (**b**) Administration of formulations generated from combinations of CpG ODN 2007, ABD2, and Poly I:C (trial 2). (**c**) Administration of formulations generated from combinations of different concentrations of CpG ODN 2007, poly I:C, and CPZ75B (trial 3).

**Table 1 vetsci-09-00203-t001:** In ovo dosage levels and protection at different time points (3, 5, and 7 days post-challenge) following an *E. coli* challenge of stand-alone immunostimulants (n = 20 embryos/group).

Group	Treatment	Dosage/Embryo	Survival Proportion (%)		
			3 Days PC	5 Days PC	7 Days PC
A	CpG ODN2007	50 µg	86	81	76
		20 µg	81	81	81
		10 µg	81	77	77
		05 µg	60	60	60
		01 µg	52	48	44
B	Poly I:C	20 µg	82	82	82
		10 µg	78	78	65
		05 µg	51	51	51
C	Cyclic Polyphosphazene (CPZ) 75B	20 µg	60	60	60
		10 µg	56	44	39
D	PBS		64	64	64

PC: post-challenge.

**Table 2 vetsci-09-00203-t002:** Trial 2—Single use and combination of effective innate immune stimulants (n = 50 embryos/group).

Group	Treatment	Dosage/Embryo	Survival % Age		
			3 Days PC	5 Days PC	7 Days PC
A	CpG	20 µg	92	85	77
B	Poly I:C	20 µg	82	82	82
C	ABD2	10 µg	95	89	84
D	CpG + Poly I:C	10 µg + 15 µg	100	100	100
E	Poly I:C + ABD2	15 µg + 10 µg	85	85	79
F	CpG + Poly I:C + ABD2	10 µg + 15 µg + 10 µg	82	82	70
G	PBS		62	62	59

**Table 3 vetsci-09-00203-t003:** Trial 3—Further study for combinations of effective innate immune stimulants (n = 30 embryos/group).

Group	Treatment	Dosage/Embryo	Survival % Age		
			3 Days PC	5 Days PC	7 Days PC
A	CpG + CPZ75B	20 µg + 10 µg	97	92	82
B	Poly I:C + CPZ75B	10 µg + 10 µg	77	75	60
C	CpG + Poly I:C + CPZ75B	20 µg + 10 µg + 10 µg	97	95	85
D	PBS		85	72	62

## Data Availability

Not applicable.

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
