# Peer review of "Characterization of Dosage Levels for In Ovo Administration of Innate Immune Stimulants for Prevention of Yolk Sac Infection in Chicks"

_vetsci, 2022, doi:10.3390/vetsci9050203_

Round 1

Reviewer 1 Report

Dear authors,

In this paper you present the application of various substances that stimulate the immune response via in ovo application in chicken embryos.  The experiment is well set up, the trials match, and together they give good results. 

Some general and specific comments:

  • you use the word reagent (L18) and later agents (L 88) - please uniform this in the whole text.
  • L36-38- this sentence is a bit to long and confusing, please reformulate it
  • L 50-52 - the sentence is also too long
  • L57 - to what you refer as "these studies"?
  • L100 - administration of treatments? can you please mention the manufacturer of the agars?
  •  L 132 - is the E. coli challenge related to Trial 1 and Trial 2? perhaps in this chapter state how sampling was performed, on which days after infection?
  •  L 189 - title could be changed (protection of combination?)
  • L 237 Discussion - the discussion is almost entirely devoted to the innate immune system, and almost no mention is made of the results of trials and their similarity or difference with other research on this topic (a bit in L 336-346). Most of the Discussion concerns selected agents and their impact on the innate immune system, and this has been in little discussed in the Introduction. it is necessary to reconcile Introduction and Discussion, and present the results through Discussion 

Reviewer 2 Report

  • Introduction

Lines 49-50: Give some references related to the importance of using vaccines against colibacillosis (Koutsianos Dimitris, Gantelet Hubert, Franzo Giovanni, Lecoupeur Mathilde, Thibault Eric, Cecchinato Mattia, Koutoulis Konstantinos C (2020) An assessment of the level of protection against colibacillosis conferred by several autogenous and/or commercial vaccination programs in conventional pullets upon experimental challenge. Veterinary Sciences 7 (3) 80). In my opinion, it is a tool which will be used in the future more and more and cannot be excluded from the discussion.  

  • Materials & Methods

Lines 93-95: Changed fonds. It is different than the rest of the text.

Lines 133-134: What was the serotype of the E.coli challenge strain? What was its origin? Was it taken from a poultry sample? What was the lesion origin? Please give more details.

Lines 134-135: How did you assess its pathogenicity  before challenge? Did you make pathogenicity tests or have you tested for virulence factors? And if not, why?

  • Results

In the materials and methods section you mentioned that you had made an enumeration of E.coli pathogens after challenge. Do you have any results concerning E.coli counts on different innate immune stimulants treatments?

  • Discussion

You have mentioned that in-ovo administration of innate immune stimulants for prevention of ECM is safe and effective. What about the legislation? Is it allowed to use such substances in poultry production (including EU legislation?). Also, it would be good to give a reference about the importance of a good in-ovo vaccination.

  • You have mentioned that in literature innate immune stimulants have been reported to enhance immune activation. Have you tried to assess the protection against yolk sack infection achieved by the administration of innate immune stimulants through investigation of the blood cell changes?
